The in vitro and in vivo effects of constitutive light expression on a bioluminescent strain of the mouse enteropathogen Citrobacter rodentium

Read Hannah M. 1 2
Mills Grant 1 2
Johnson Sarah 1 2
Tsai Peter 2 3
Dalton James 1 2 4
Barquist Lars 5
Print Cristin G. 2 3 4
Patrick Wayne M. 4 6
Wiles Siouxsie s.wiles@auckland.ac.nz 1 2 4
1 Bioluminescent Superbugs Lab, University of Auckland , Auckland , New Zealand
2 Department of Molecular Medicine and Pathology, University of Auckland , Auckland , New Zealand
3 Bioinformatics Institute, School of Biological Sciences, University of Auckland , Auckland , New Zealand
4 Maurice Wilkins Centre for Molecular Biodiscovery , New Zealand
5 Institute for Molecular Infection Biology, University of Würzburg , Würzburg , Germany
6 Department of Biochemistry, University of Otago , Dunedin , New Zealand
O’Byrne Conor
Electronic publication date: 2016 Jun 22
Publication date: 2016
Volume: 4
Electronic Location ID: e2130
Received 2016 Apr 22; Accepted 2016 May 24
Copyright: ©2016 Read et al.
Copyright year: 2016
Copyright holder: Read et al.
License: This is an open access article distributed under the terms of the Creative Commons Attribution License, which permits unrestricted use, distribution, reproduction and adaptation in any medium and for any purpose provided that it is properly attributed. For attribution, the original author(s), title, publication source (PeerJ) and either DOI or URL of the article must be cited.
License URL: https://creativecommons.org/licenses/by/4.0/

Keywords: Bioluminescence, Lux, Luciferase, Biophotonic imaging, Bioluminescence imaging, Enteric pathogens, Animal model, Reporter genes, Phenotypic microarray, Biolog

Funding: Maurice Wilkins Centre for Molecular Biodiscovery Health Research Council of New Zealand SW (09/099) Alexander von Humboldt Stiftung/Foundation This work was supported by seed funding from the Maurice Wilkins Centre for Molecular Biodiscovery, and by a Sir Charles Hercus Fellowship to SW (09/099) from the Health Research Council of New Zealand. LB is supported by a Research Fellowship from the Alexander von Humboldt Stiftung/Foundation. The funders had no role in study design, data collection and analysis, decision to publish, or preparation of the manuscript.

==============================
Bioluminescent reporter genes, such as those from fireflies and bacteria, let researchers use light production as a non-invasive and non-destructive surrogate measure of microbial numbers in a wide variety of environments. As bioluminescence needs microbial metabolites, tagging microorganisms with luciferases means only live metabolically active cells are detected. Despite the wide use of bioluminescent reporter genes, very little is known about the impact of continuous (also called constitutive) light expression on tagged bacteria. We have previously made a bioluminescent strain of Citrobacter rodentium, a bacterium which infects laboratory mice in a similar way to how enteropathogenic Escherichia coli (EPEC) and enterohaemorrhagic E. coli (EHEC) infect humans. In this study, we compared the growth of the bioluminescent C. rodentium strain ICC180 with its non-bioluminescent parent (strain ICC169) in a wide variety of environments. To understand more about the metabolic burden of expressing light, we also compared the growth profiles of the two strains under approximately 2,000 different conditions. We found that constitutive light expression in ICC180 was near-neutral in almost every non-toxic environment tested. However, we also found that the non-bioluminescent parent strain has a competitive advantage over ICC180 during infection of adult mice, although this was not enough for ICC180 to be completely outcompeted. In conclusion, our data suggest that constitutive light expression is not metabolically costly to C. rodentium and supports the view that bioluminescent versions of microbes can be used as a substitute for their non-bioluminescent parents to study bacterial behaviour in a wide variety of environments.

Introduction

Bioluminescence is the by-product of a chemical reaction which has evolved in a wide variety of creatures for different purposes. This ‘living light’ allows fireflies like Photinus pyralis to find a mate (Vencl, 2004), larvae like the New Zealand glow worm Arachnocampa luminosa to lure prey (Meyer-Rochow, 2007), and the bacterium Aliivibrio fischeri (formally Vibrio fischeri) to camouflage its nocturnal symbiont, the Hawaiian bobtail squid, while hunting (Jones & Nishiguchi, 2004). Bioluminescence is produced by the oxidation of a substrate (a luciferin) by an enzyme (a luciferase), which usually requires energy and oxygen. Cloning of the bioluminescence genes from P. pyralis (De Wet et al., 1985), V. fischeri (Engebrecht, Nealson & Silverman, 1983) and Photorhabdus luminescens (Szittner & Meighen, 1990), has let researchers use light production as a real-time non-invasive and non-destructive surrogate measure of microbial numbers in a wide variety of different culture environments, including within laboratory animals (Andreu, Zelmer & Wiles, 2011). This has proven particularly useful for studying microorganisms which take several weeks to grow on selective media, such as the bacterium Mycobacterium tuberculosis (Andreu et al., 2012; Andreu et al., 2013). As bioluminescence requires microbial metabolites, such as ATP and reduced flavin mononucleotide (FMNH2), tagging microorganisms with luciferases means only live, metabolically active cells are detected.

Of the available bioluminescent reporter systems, the most widely used in bacteriology research is the bacterial luminescence reaction, encoded by the lux gene operon. The reaction involves the oxidation of a long chain aldehyde and FMNH2, resulting in the production of oxidised flavin (FMN), a long chain fatty acid, and the emission of light at 490 nm (Hastings, 1978). The reaction is catalysed by bacterial luciferase, a 77 kDa enzyme made up of an alpha and a beta subunit encoded by the luxA and luxB genes, respectively. The luxC, D and E genes encode the subunits of a multi-enzyme complex responsible for regenerating the aldehyde substrate from the fatty acid produced by the reaction. A significant advantage of the bacterial bioluminescence system is the ability to express the biosynthetic enzymes for substrate synthesis, allowing light to be produced constitutively. One of the underlying motivations for using lux-tagged bacteria is the reduction in the number of animals needed for in vivo experiments, an ethical and legislative requirement in many countries. Using a technique known as biophotonic imaging, tagged bacteria can be non-invasively and non-destructively visualised and quantified on multiple occasions from within the same group of infected animals, whereas culture based techniques need groups of animals to be euthanised at each time point of interest (Andreu, Zelmer & Wiles, 2011). However, very little is known about the impact of constitutive light expression on tagged bacteria. We hypothesise that light production will impose a metabolic burden on the tagged bacteria, with the actual fitness costs dependent on the host bacterial species, the site of insertion of the bioluminescence genes and their expression levels.

We have previously made a lux-tagged derivative of Citrobacter rodentium (Wiles et al., 2005), a bacterium that infects laboratory mice using the same virulence mechanisms as the life-threatening pathogens, enteropathogenic Escherichia coli (EPEC) and enterohaemorrhagic E. coli (EHEC) use to infect humans (Mundy et al., 2005; Collins et al., 2014). C. rodentium ICC180 contains a single chromosomally-located copy of the lux operon from P. luminescens, alongside a gene for resistance to the antibiotic kanamycin. We have previously non-invasively tracked ICC180 during infection of mice (Wiles et al., 2006), demonstrating that C. rodentium rapidly spreads between infected and uninfected animals and that bacteria shed from infected mice are 1,000 times more infectious than laboratory grown bacteria (Wiles, Dougan & Frankel, 2005). While we have shown that ICC180 can reach similar numbers within the gastro-intestinal tracts of infected mice and causes similar pathology when compared to its non-bioluminescent parent strain ICC169 (Wiles et al., 2005), we have never fully investigated the impact of constitutive light expression on the fitness of ICC180.

In this study we set out to determine whether C. rodentium ICC180 has a competitive disadvantage when competed against its non-bioluminescent parent ICC169 in a range of in vitro and in vivo environments. We also sequenced the genome and associated plasmids of ICC180 to determine whether there were any other genetic differences between the two strains, perhaps as a result of the transposon mutagenesis technique (Winson et al., 1998) used to generate ICC180. Finally, we compared the growth profiles of the two strains using the BIOLOG Phenotypic Microarray (PM) system, a rapid 96-well microtitre plate assay for phenotypically profiling microorganisms based on their growth under approximately 2,000 different metabolic conditions (Bochner, Gadzinski & Panomitros, 2001).

Materials and methods

Bacterial strains and culture conditions

The bacterial strains used in this study were Citrobacter rodentium ICC169 (spontaneous nalidixic acid resistant mutant) (Wiles et al., 2005) and ICC180 (nalidixic acid and kanamycin resistant) (Wiles et al., 2005). Bacteria were revived and grown from frozen stocks stored at –80°C in order to prevent adaptation of C. rodentium over multiple laboratory subcultures. Bacteria were grown at 37°C with shaking at 200 revolutions per minute (RPM) in LB-Lennox media (Fort Richard Laboratories Ltd., Auckland, New Zealand) or in defined minimal media (modified Davis & Mingioli media (Davis, 1949)), containing ammonium sulphate [1 g l−1], potassium dihydrogen phosphate [4.5 g l−1], dipotassium hydrogen phosphate anhydrous [10.5 g l−1], sodium citrate dihydrate [5 g l−1], magnesium sulfate heptahydrate [24.65 mg l−1], thiamine [0.5 mg l−1], supplemented with 1% glucose) at 37°C. Antibiotics (kanamycin [50 ug ml−1], nalidixic acid [50 ug ml−1]) were only added to the media if they were required for selection. All chemicals and antibiotics were obtained from Sigma-Aldrich (Castle Hill NSW, Australia).

Genome sequencing and analysis

Genomic DNA was prepared from bacteria grown overnight in LB-Lennox broth. Whole genome sequencing was performed using the Illumina HiSeq platform by BGI (Hong Kong). A total of 3,414,820 paired-end 90 bp reads were generated for ICC169 and 3,369,194 for ICC180. Data was quality trimmed using DynamicTrim (Cox, Peterson & Biggs, 2010) (minimum Phred score 25) and filtering of reads shorter than 45 bp after quality trimming was performed using LengthSort (Cox, Peterson & Biggs, 2010); both programmes are part of the SolexaQA software package (Cox, Peterson & Biggs, 2010). After filtering, 2,444,336 paired reads were retained for ICC169 and 2,383,491 for ICC180. All remaining high quality and properly paired reads were mapped to the reference strain C. rodentium ICC168 (Genbank accession number FN543502.1(Petty et al., 2010)) using the default settings in BWA (Li & Durbin, 2010). On average, 95% of all high quality reads mapped uniquely to ICC168 (94.8% for ICC169 and 95.2% for ICC180) and single nucleotide polymorphisms (SNPs) and indels that were present only in ICC180 at 100% were identified using Samtools mpileup (Li et al., 2009). SNPs and indels were confirmed by PCR and sequencing. In addition, the reads were also analysed using BreSeq version 0.24rc6 (Deatherage & Barrick, 2014), which identified predicted mutations that were statistically valid. To locate the insertion site of the lux operon and kanamycin resistance (KmR) gene, we first performed de novo assembly on quality trimmed data for ICC180 using EDENA v3.0 (Hernandez et al., 2008). All assembled contigs were mapped to the C. rodentium reference strain ICC168 using Geneious (Kearse et al., 2012) and contigs unmapped to ICC168 were BLAST searched against the lux operon and KmR gene. We located both the lux operon and KmR gene on an unmapped contig 117,921 bp long. To identify the position of this contig, we broke the contig into two segments based on the location of lux operon and KmR gene positions on the contig, and performed additional reference mapping to ICC168 to identify the insertion site. To determine changes to the plasmids present in C. rodentium, reads were also mapped to the sequenced plasmids pCROD1 (Genbank accession number FN543503.1), pCROD2 (Genbank accession number FN543504.1), pCROD3 (Genbank accession number FN543505.1), and pCRP3 (Genbank accession number NC_003114).

Phenotypic microarrays

Phenotypic microarrays were performed by BIOLOG Inc. (California, USA) as described previously (Bochner, Gadzinski & Panomitros, 2001). Assays were performed in duplicate using plates PM1-20 (Table S1). The data was exported and analysed in the software package R as previously described (Reuter et al., 2014). Briefly, growth curves were transformed into Signal Values (SVs) (Homann et al., 2005) summarising the growth over time while correcting for background signal. Principal component analyses showed a clear separation by genotype, suggesting reproducible differences in metabolism between the two strains. A histogram of log signal values displayed a clear bimodal distribution, which we interpreted as representing non-respiring cells (‘off’, low SV) and respiring cells (‘on’, high SV), respectively. Normal distributions were fitted to these two distributions using the R MASS package, and these models were then used to compute log-odds ratios for each well describing the probability that each observation originated from the ‘on’ or ‘off’ distribution. Wells which were at least 4 times more likely to come from the ‘on’ distribution than the ‘off’ in both replicates were considered to be actively respiring. In order to determine the significance of observed differences between genotypes, we applied the moderated t-test implemented in the limma R/Bioconductor package (Smyth, 2004). Wells with a Benjamini–Hochberg corrected P-value of less than 0.05, that is allowing for a false discovery rate of 5%, and which were called as actively respiring for at least one genotype, were retained for further analysis. The data was also analysed using the DuctApe software suite (Galardini et al., 2014). Growth curves were analysed using the dphenome module, with the background signal subtracted from each well. Based on the results of an elbow test (Fig. S1), 7 clusters were chosen for k-means clustering. An Activity Index (AV) was created based on the clustering, ranging from 0 (minimal activity) to 6 (maximal activity). AV data was visualised using the plot and ring commands of the dphenome module (Fig. S2).

In vitro growth experiments

Briefly, for individual growth curves, 10 ml of either LB-Lennox or defined minimal medium was inoculated with 20 µl of a culture grown overnight in LB-Lennox broth. Cultures were grown at 37°C with shaking at 200 RPM and samples were removed at regular intervals to measure bioluminescence, using a VICTOR X Light Plate reader (Perkin Elmer), and viable counts, by plating onto LB-Lennox Agar (Fort Richard Laboratories Ltd., Auckland, New Zealand). Overnight cultures were plated to determine the initial inocula. Experiments were performed on seven separate occasions and results used to calculate area under curve (AUC) values for each strain. For the competition experiments, 10 µl of a culture grown overnight in LB-Lennox broth was used to inoculate 1 ml of defined minimal medium, with the mixed culture tubes receiving 5 µl of each strain. Inoculated tubes were incubated overnight at 37°C with shaking at 200 RPM, followed by serial dilution in sterile phosphate buffered saline (PBS) for plating onto LB Agar containing either nalidixic acid or kanamycin. The ratio of colonies that grew on media containing each antibiotic was used to determine the proportion of each strain remaining. Experiments were performed on eight separate occasions and the results used to calculate AUC values and competitive indices (CI). CI’s were calculated as follows: CI = [strain of interest output/competing strain output]/[strain of interest input/competing strain input] (Freter, O’Brien & Macsai, 1981; Taylor et al., 1987).

Infection of Galleria mellonella

5th instar Galleria mellonella larvae (waxworms) were obtained from a commercial supplier (Biosuppliers.com, Auckland, New Zealand). Bacteria were grown overnight in LB-Lennox broth and used to infect waxworms which were pale in colour and weighed approximately 100–200 mg. Waxworms were injected into one of the last set of prolegs with 20 µl of approximately 108 colony forming units (CFU) of bacteria using a 1ml fine needle insulin syringe. Waxworms were injected with either ICC169, ICC180 or a 1:1 mix and incubated at 37°C. Throughout the course of a 24 h infection, individual waxworms were inspected for phenotypic changes and scored using a standardised method for assessing waxworm health (the Caterpillar Health Index [CHI]) which we have developed. Briefly, waxworms were monitored for movement, cocoon formation, melanisation, and survival. Together, these data form a numerical scale, with lower CHI scores corresponding with more serious infections and higher scores with healthier waxworms. Scores were used to calculate AUC values. Bioluminescence (given as relative light units [RLU]) was measured at regular intervals from waxworms infected with ICC180 (Fig. S3). Waxworms were placed into individual wells of a dark OptiPlate-96 well microtitre plate (Perkin Elmer) and bioluminescence measured for 1 s to provide relative light units (RLU)/second using the VICTOR X Light Plate reader. Waxworms infected with ICC169 were used as a control. Following death, or at 24 h, waxworms were homogenised in PBS and plated onto LB-Lennox Agar containing the appropriate antibiotics. Independent experiments were performed three times using 10 waxworms per group.

Infection of Mice

Female 6–7 week old C57BL/6Elite mice were provided by the Vernon Jansen Unit (University of Auckland) from specific-pathogen free (SPF) stocks. All animals were housed in individually HEPA-filtered cages with sterile bedding and free access to sterilised food water. Experiments were performed in accordance with the New Zealand Animal Welfare Act (1999) and institutional guidelines provided by the University of Auckland Animal Ethics Committee, which reviewed and approved these experiments under applications R1003 and R1496. Bacteria grown overnight in LB-Lennox broth were spun at 4500 RPM for 5 min, and resuspended in a tenth of the volume of sterile PBS, producing a 10x concentrated inoculum. Animals were orally inoculated using a gavage needles with 200 µl of either ICC169, ICC180, or a 1:1 mix (containing approximately 108 CFU of bacteria) and biophotonic imaging used to determine correct delivery of bacteria to the stomach. The number of viable bacteria used as an inoculum was determined by retrospective plating onto LB-Lennox Agar containing either nalidixic acid or kanamycin. Stool samples were recovered aseptically at various time points after inoculation, and the number of viable bacteria per gram of stool was determined after homogenisation at 0.1 g ml−1 in PBS and plating onto LB-Lennox Agar containing the appropriate antibiotics. The number and ratio of colonies growing on each antibiotic was used to calculate AUC values and CI’s as described above. Independent experiments were performed twice using 6 animals per group.

In vivo bioluminescence imaging

Biophotonic imaging was used to noninvasively measure the bioluminescent signal emitted by C. rodentium ICC180 from anaesthetised mice to provide information regarding the localisation of the bacterium. Prior to being imaged, the abdominal area of each mouse was shaved, using a Vidal Sasoon handheld facial hair trimmer, to minimise any potential signal impedance by melanin within pigmented skin and fur. Mice were anaesthetised with gaseous isoflurane and bioluminescence (given as photons s−1 cm−2 steradian [sr]−1) was measured using the IVIS® Kinetic camera system (Perkin Elmer). A photograph (reference image) was taken under low illumination before quantification of photons emitted from ICC180 at a binning of four over 1 min using the Living Image software (Perkin Elmer). The sample shelf was set to position D (field of view, 12.5 cm). For anatomic localisation, a pseudocolor image representing light intensity (blue, least intense to red, most intense) was generated using the Living Image software and superimposed over the gray-scale reference image. Bioluminescence in specific regions of individual mice also was quantified using the region of interest tool in the Living Image software program (given as photons s−1) and used to calculate AUC values for each individual animal.

Statistical analyses

Data was analysed using GraphPad Prism 6. Data was tested for normality using the D’Agostino-Pearson test; data which failed normality was analysed using a non-parametric test, while data which passed normality was analysed using a parametric test. One-tailed tests were used to test the hypothesis that constitutively expressing light gives ICC180 a differential fitness cost compared to the non-bioluminescent parent strain ICC169. When comparing multiple experimental groups, Dunn’s post hoc multiple comparison test was applied.

Results

Bioluminescent Citrobacter rodentium strain ICC180 has three altered chromosomal genes and a large deletion in plasmid pCROD1 in addition to insertion of the lux operon and kanamycin resistance gene

We determined the whole genome draft sequences of C. rodentium ICC169 and ICC180 using Illumina sequence data. Compared with sequenced type strain ICC168 (Genbank accession number FN543502.1), both strains have a substitution of a guanine (G) to an adenine (A) residue at 2,475,894 bp, resulting in an amino acid change from serine (Ser) to phenylalanine (Phe) within gyrA, the DNA gyrase subunit, and conferring resistance to nalidixic acid. The sequencing data indicate that the lux operon and kanamycin resistance gene (a 7,759 bp fragment) has inserted at 5,212,273 bp, disrupting the coding region of a putative site-specific DNA recombinase (Fig. 1). In addition to the presence of the lux operon and kanamycin resistance gene, we found that the genome of ICC180 differs from ICC169 by two single nucleotide polymorphisms (SNPs), a single base pair insertion (of a G residue at 3,326,092 bp which results in a frameshift mutation within ROD_31611, a putative membrane transporter) and a 90 bp deletion in deoR (deoxyribose operon repressor) (Table 1). All four plasmids previously described for C. rodentium were present in ICC180, however the largest of these plasmids, pCROD1, shows evidence of extensive deletion events and is missing 41 out of 60 genes (Table S2).

Figure 1 Whole genome sequencing shows that the lux operon and kanamcyin resistance gene have inserted at position 5,212,273 in the chromosome of C. rodentium ICC180, disrupting a putative site-specific DNA recombinase.

Table 1 SNPs and indels that differ between the bioluminescent C. rodentium derivative ICC180 and its parent strain ICC169.

Sequencing revealed three points of difference between ICC180 and ICC169. Two SNPs are present, each cytosine substitutions, and one guanine insertion inducing a frameshift mutation. Sequencing data was analysed using BreSeq (Deatherage & Barrick, 2014).

Position	Base change	Amino acid change	Gene	Function	
2,936,285	T→C	D471G (GAC→GGC)	cts1V	T6SS protein Cts1V	
3,999,002	T→C	E89G (GAG→GGG)	pflD	Formate acetyltransferase 2	
3,326,092	CAG→CAGG	Frameshift	ROD˙31611	Major Facilitator Superfamily transporter	

Table 2 Phenotypic microarray (PM) wells in which the growth of bioluminescent C. rodentium derivative ICC180 significantly differs from its non-bioluminescent parent strain ICC169.

PM class	Substrate	Adjusted P value	Improved growth by ICC169	Improved growth by ICC180	Comment	
Nitrogen	D-glucosamine	0.0159		√		
	Cytidine	0.0280		√		
	Ala-His	0.0316		√		
Phosphate	Inositol hexaphosphate	0.0280		√		
Nitrogen peptides	Lys-Asp	0.0306	√			
Chemicals	Kanamycin	0.0076		√	Conferred by KanR gene	
	Paromomycin	0.0048		√	Aminoglycoside—the kanamycin cassette will be mediating resistance	
	Geneticin	0.0048		√	Aminoglycoside—the kanamycin cassette will be mediating resistance	
	Dequalinium chloride	0.0116		√	Quaternary ammonium salt	
	Spiramycin	0.0088		√	Macrolide—acts at ribosomal 50S, c.f. aminoglycosides at 30S	
	Rolitetracycline	0.0316		√	Tetracycline; prevents tRNA binding at 30S A-site	
	Doxycycline	0.0210		√	Tetracycline; prevents tRNA binding at 30S A-site	
	Coumarin	0.0333		√	Fragrant organic compound found in many plants	
	Iodonitro tetrazolium violet (INT)	0.0087		√	Electron acceptor, reduced by succinate dehydrogenase (and by superoxide radicals)	
	EDTA	0.0048	√		Metal chelator	
	EGTA	0.0210	√		Metal chelator	
	Rifampicin	0.0048	√		RNA polymerase inhibitor	
	Colistin	0.0048	√		Cyclic polypeptide; disrupts outer membrane	
	Oxycarboxin	0.0121	√		Fungicide	
	Phenethicillin	0.0048	√		Beta-lactam	
	Cytosine-1-b-D-arabinofuranoside	0.0123	√		Nucleoside analogue (anti-cancer/-viral)	
	Sodium Nitrate	0.0306	√			
	Cefoxitin	0.0316		√	Beta-lactam	
	Disulphiram	0.0349		√	Inhibits acetaldehyde dehydrogenase	

Constitutive light expression does not greatly impact the metabolism of C. rodentium ICC180

C. rodentium ICC169 and its bioluminescent derivative ICC180 were grown on two separate occasions using PM plates 1–20. We analysed the data using the DuctApe software suite which calculates an activity index (AV) for each strain in response to each well (Fig. S2). Next, the growth curve data were transformed into Signal Values (SVs) as previously described (Reuter et al., 2014), summarising the growth of each strain over time for each well. Wells which were considered to be actively respiring were analysed using the moderated t-test implemented in the limma R/Bioconductor package (Smyth, 2004). Those wells with a Benjamini–Hochberg corrected P-value of less than 0.05 are shown in Table 2 (with corresponding growth curves in Fig. 3). Our results indicate that the growth of the two strains significantly differed (p = < 0.05) in 26/1,920 wells. Of these >80% are from the PM11-20 plates, which belong to the chemical category, suggesting that the expression of bioluminescence is near-neutral in almost every non-toxic environment. The bioluminescent strain ICC180 is able to use D-glucosamine, cytidine and Ala-His as nitrogen sources, and inositol hexaphosphate as a phosphate source, and grew significantly better than ICC169 in the presence of 11 chemicals: the antibiotics kanamycin, paromomycin, geneticin, spiramycin, rolitetracycline, doxycycline, cefoxitin; the quaternary ammonium salt dequalinium chloride; coumarin; iodonitrotetrazolium violet; and the acetaldehyde dehydrogenase inhibitor disulphiram (Table 2). That the expression of a kanamycin resistance gene also improves growth of ICC180 in the presence of related aminoglycosides is reassuring. In contrast, the wildtype strain ICC169 was able to use the nitrogen peptide Lys-Asp and grew significantly better in the presence of 8 chemicals: the metal chelators, EDTA and EGTA, sodium nitrate, the antibiotics rifampicin and phenethicillin, the fungicide oxycarboxin, the cyclic polypeptide colistin, the nucleoside analogue cytosine-1-b-D-arabinofuranoside and (Table 2). The fact that significant differences in growth rate were observed for so few conditions, provided robust and comprehensive evidence that light production is near-neutral in C. rodentium ICC180.

The growth of ICC180 is not impaired in a rich laboratory medium, when compared to its non-bioluminescent parent strain, but does exhibit an increased lag phase when grown in a restricted medium

We grew ICC180 and ICC169 in rich (LB-Lennox) and restricted (minimal A salts with 1% glucose supplementation) laboratory media. For ICC180, we found that bioluminescence strongly correlated with the bacterial counts recovered throughout the growth period in both LB-Lennox (Spearman’s r = 0.9293 95% CI [0.8828–0.9578], p = < 0.0001) and the restricted medium (Spearman’s r = 0.9440 95% CI [0.9001–0.9689], p = < 0.0001) (Figs. 2A, 2B, 3A &3B). We also found that the growth of each strain was comparable in LB-Lennox medium, with no significant difference between the bacterial counts recovered over 8 h (Fig. 2B), as demonstrated by the calculated AUC values (Fig. 2C).

Figure 2 C. rodentium ICC180 is not impaired during growth in a rich laboratory medium when compared to its non-bioluminescent parent strain ICC169.

Wildtype C. rodentium ICC169 (shown as purple circles) and its bioluminescent derivative ICC180 (shown as blue triangles) were grown in LB-Lennox broth and monitored for changes in bioluminescence (given as relative light units [RLU] ml−1) (A) and bacterial counts (given as colony forming units [CFU] ml−1) (B). Bacterial count data was used to calculate area under curve (AUC) values for each strain (C). Data (medians with ranges where appropriate) is presented from experiments performed on eight separate occasions.

Figure 3 C. rodentium ICC180 is mildly impaired during growth in a defined minimal laboratory medium when compared to its non-bioluminescent parent strain ICC169.

Wildtype C. rodentium ICC169 (shown as purple circles) and its bioluminescent derivative ICC180 (shown as blue triangles) were grown in minimal A salts supplemented with 1% glucose and monitored for changes in bioluminescence (given as relative light units [RLU] ml−1) (A) and bacterial counts (given as colony forming units [CFU] ml−1) (B). Bacterial count data was used to calculate area under curve (AUC) values for each strain, which were found to be significantly different (p = 0.0078; Wilcoxon Matched pairs-signed rank test) (C). Data (medians with ranges where appropriate) is presented from experiments performed on eight separate occasions.

We also found no significant difference between the bacterial counts recovered from ICC180 and ICC169 growing in the restricted medium for 14 h (mean CFU 5.67 × 108 [SD 2.31 × 108] and 8.84 × 108 [SD 2.93 × 108], respectively). However, we did find a significant difference between the AUC values calculated from the bacterial counts recovered over the course of 14 h (p = 0.0078, one-tailed Wilcoxon matched-pairs signed rank test) (Fig. 3C). We calculated the slopes of the growth curves and found that there was no difference in the rates of growth of the two strains during exponential phase. Instead, we found a significant difference between the slopes calculated during the first 4 h of growth (1/slope values: ICC169 =1.48 × 10−7 [SD 9.98 × 10−8], ICC180 =2.47 × 10−7 [SD 1.10 × 10−7]; p = 0.0041, one-tailed Paired t test), suggesting ICC180 spends longer in lag phase than ICC169 when grown in the restricted medium used.

ICC180 is not impaired in the Galleria mellonella infection model

We infected larvae of the Greater Wax Moth G. mellonella (waxworms) with ICC169 and ICC180 in single and 1:1 mixed infections. We monitored the waxworms over a 24–48 h period for survival and disease symptoms. The Caterpillar Health Index (CHI) is a numerical scoring system which measures degree of melanisation, silk production, motility, and mortality. We found that the majority of infected waxworms succumb to C. rodentium infection (Fig. 4A), which is reflected by the concurrent decrease in CHI score (Fig. 4B). This is in contrast to waxworms injected with PBS, who all survived and consistently scored 9–10 on the CHI scale throughout the experiments. We also found that the survival and symptoms of waxworms infected with each strain were comparable, with no significant difference between the survival curves (Fig. 4A), and calculated AUC values for the CHI scores (Fig. 4C). However, when we directly compared ICC169 and ICC180 in mixed infections of approximately 1:1, we found a significant difference in the relative abundance of the bacteria recovered from waxworms at either time of death or 24 h, whichever occurred first (p = 0.001, one-tailed Wilcoxon matched-pairs signed rank test). Despite a slightly lower infectious dose, higher numbers of ICC180 were consistently recovered from infected waxworms (Fig. 4D).

Figure 4 Bioluminescent C. rodentium ICC180 is not impaired in the Galleria mellonella infection model.

Groups of larvae (n = 10) of the Greater Wax Moth Galleria mellonella were infected with ICC169 and ICC180 in single and 1:1 mixed infections and monitored for survival (%) (A) and for disease symptoms using the Caterpillar Health Index (CHI), a numerical scoring system which measures degree of melanisation, silk production, motility, and mortality (given as median CHI values) (B). Survival curves (A) and calculated area under curve (AUC) data of CHI scores reveals no difference between waxworm response to infection from either strain (C). Waxworms infected with a 1:1 mix of ICC169 and ICC180 were homogenised at 24-hours, or at time of death if earlier. Actual infecting doses for each strain were determined by retrospective plating, and are indicated by *. The bacterial burden of ICC180 and ICC169 in individual caterpillars (indicated by the dotted line), was calculated after plating onto differential media and found to be significantly different (p = 0.001; one-tailed Wilcoxon matched pairs-signed rank test) (D). Data (medians with ranges where appropriate) is presented from experiments performed on three separate occasions, except (A) and (D), where the results of a representative experiment are shown.

ICC180 is impaired in mixed but not in single infections in mice when compared to its non-bioluminescent parent strain

We orally gavaged groups of female 6–8 week old C57Bl/6 mice (n = 6) with ∼5 × 109 CFU of ICC169 and ICC180, either individually or with a 1:1 ratio of each strain. We followed the infection dynamics by obtaining bacterial counts from stool samples (Fig. 5) and by monitoring bioluminescence from ICC180 using biophotonic imaging (Fig. 6). We found that the growth of each strain was comparable during single infections, with no significant difference between the bacterial counts recovered throughout the infection (Fig. 5A), as demonstrated by the calculated AUC values (Fig. 5B).

Figure 5 C. rodentium ICC180 is impaired during mixed, but not in single, infections in mice when compared to its non-bioluminescent parent strain ICC169.

Groups of female 6–8 week old C57Bl/6 mice (n = 6) were orally-gavaged with ∼5 × 109 CFU of wildtype C. rodentium ICC169 (shown as purple circles) and its bioluminescent derivative ICC180 (shown as blue triangles) in single infections (A, B) or 1:1 mixed infections (C, D) and monitored for changes in bacterial counts (given as colony forming units [CFU] g−1 stool) (A, B). Bacterial count data was used to calculate area under curve (AUC) values for each strain in single (B) and mixed (D) infections, and were found to be significantly different only for the mixed infections (p = 0.001; one-tailed Wilcoxon Matched pairs-signed rank test). This is reflected in the competitive indices (CI) calculated from the bacterial counts recovered during mixed infections, with ICC180 showing a growing competitive disadvantage from day 2 post-infection (E). Data (medians with ranges where appropriate) is presented from experiments performed on two separate occasions.

Figure 6 Despite having a fitness disadvantage in mixed infections of mice, ICC180 is still visible by biophotonic imaging.

Groups of female 6–8 week old C57Bl/6 mice (n = 6) were orally-gavaged with ∼5 × 109 CFU of wildtype C. rodentium ICC169 and its bioluminescent derivative ICC180 in single infections or 1:1 mixed infections. Mice were anaesthetised with gaseous isoflurane and bioluminescence (given as photons s−1 cm−2 sr−1) from ICC180 measured using the IVIS® Kinetic camera system (Perkin Elmer). The images show changes in peak bioluminescence over time with variations in colour representing light intensity at a given location and superimposed over a grey-scale reference image (A). Red represents the most intense light emission, whereas blue corresponds to the weakest signal. Bioluminescence from the abdominal region of individual mice was quantified (as photons s−1) using the region of interest tool in the Living Image software program and used to calculate area under curve (AUC) values for each individual animal over the course of the infection (B). Dotted line represents background. Experiments were performed on two separate occasions. Three representative animals are shown; no light was detected from animals infected with ICC169 alone, while lower levels of light were detected from animals infected with a mix of ICC169 and ICC180.

In contrast, we found a significant difference between the AUC values calculated from the bacterial counts recovered from ICC180 and ICC169 during mixed infections (p = 0.001, one-tailed Wilcoxon matched-pairs signed rank test) (Fig. 5D). Our data demonstrates that when in direct competition with ICC169, ICC180 is shed at consistently lower numbers from infected animals (Fig. 5C). At the peak of infection (days 6–8), this equates to over a 10-fold difference, with mice shedding a median of 1.195 × 108 CFU (SD 4.544 × 107) for ICC169 compared to 9.98 × 106 CFU (SD 1.544 × 107) for ICC180. This disadvantage is reflected in the Competitive Indices we calculated from bacterial counts recovered at each time point, which for ICC180 decreased steadily throughout the course of the infection (Fig. 5E). Despite this disadvantage, ICC180 is never completely outcompeted and remains detectable in the stools of infected animals until the clearance of infection (Fig. 5C), and by biophotonic imaging until day 10–13 post-infection (Fig. 6A).

Discussion

Bioluminescently-labelled bacteria have gained popularity as a powerful tool for investigating microbial pathogenicity in vivo, and for preclinical drug and vaccine development (Steinhuber et al., 2008; Massey et al., 2011; Sun et al., 2012; Kassem et al., 2016). Individual infected and/or treated animals can be followed over time, in contrast to the large numbers of animals that are euthanised at specific time points of interest for quantifying bacterial loads using labour-intensive plate count methods. Most widely used is the lux operon of the terrestrial bacterium P. luminescens, which encodes for the luciferase enzyme which catalyses the bioluminescence reaction, and for a multi-enzyme complex responsible for regenerating the required substrate. As FMNH2 is also required for light production, it is generally hypothesised that light production is likely to impose a metabolic burden on tagged bacteria.

The impact of expression of the lux operon has been reported for a number of microbial species. Sanz and colleagues (2008) created strains of Bacillus anthracis that emit light during germination, by introducing plasmids with lux operon expression driven by the sspB promoter. The authors noted that the bioluminescent strains were less efficient at germinating, resulting in an increase in the dose required to cause a lethal infection in mice inoculated by either the subcutaneous or intranasal route. Despite the reduced virulence, bioluminescent B. anthracis was still capable of successfully mounting an infection, and the use of biophotonic imaging revealed new infection niches which would have been difficult to accurately measure using traditional plating methods. Similarly, a clinical M75 isolate of Streptococcus pyogenes with the lux operon chromosomally inserted at the spy0535 gene was found to have significantly attenuated maximal growth in vitro, as well as reduced survival in an intranasal mouse model (Alam et al., 2013). The bioluminescent Listeria monocytogenes Xen32 strain was shown to cause reduced mortality after oral inoculation of BALB/cJ mice, however subsequent investigation revealed that the chromosomally-located lux operon had inserted into the flaA gene, disrupting the ability of Xen32 to produce flagella. This suggests that the virulence attenuation observed is likely due to the location of the lux operon rather than the metabolic cost of light production (Bergmann et al., 2013).

In this study, we compared a bioluminescent-derivative of the mouse enteropathogen C. rodentium, strain ICC180, with its non-bioluminescent parent strain ICC169, using the BIOLOG Phenotypic Microarray (PM) system, which tests microbial growth under approximately 2,000 different metabolic conditions. Rather surprisingly, our results demonstrated that the expression of bioluminescence in ICC180 is near-neutral in almost every non-toxic environment tested, suggesting that light production is not metabolically costly to C. rodentium. This supports the “free lunch hypothesis” proposed by Falls and colleagues (2014), namely that cells have an excess of metabolic power available to them. Interestingly, ICC180 grew significantly better than its non-bioluminescent parent strain in the presence of a number of different chemicals, including several antibiotics, supporting previous findings that bacteria have many pleiotropic ways to resist toxins (Soo, Hanson-Manful & Patrick, 2011). In the case of the artificial electron acceptor iodonitrotetrazolium violet, we hypothesise that light production may be altering the redox balance of the cell, thus making the dye less toxic.

We also compared the ability of ICC180 and ICC169 to directly compete with one another during infection of their natural host, laboratory mice, as well as larvae of the Greater Wax Moth G. mellonella (waxworms). Wax worms are becoming an increasingly popular surrogate host for infectious diseases studies due to legislative requirements in many countries to replace the use of animals in scientific research. Wax worms have a well-developed innate immune system involving a cellular immune response in the form of haemocytes, and a humoral immune response in the form of antimicrobial peptides in the hemolymph (Vogel et al., 2011). Detection of bacterial cell wall components leads to activation of the prophenoloxidase cascade, which is similar to the complement system in mammals (Park et al., 2005), and subsequent endocytosis of bacteria by haemocytes. The haemocytes function in a similar way to mammalian neutrophils, and kill bacteria via NADPH oxidase and production of reactive oxygen species (Bergin et al., 2005). Again, we observed no fitness costs to constitutive light production by ICC180. Interestingly, we recovered significantly more ICC180 from wax worms infected with both ICC180 and ICC169. Similar to the response to iodonitrotetrazolium violet, an altered redox balance caused by light production could make reactive oxygen species generated by the wax worm immune response, less toxic.

In contrast, our data shows that the non-bioluminescent parent strain ICC169 has a clear competitive advantage over ICC180 during infection of adult C57Bl/6 mice, with the bioluminescent strain shed from infected animals at consistently lower numbers. Surprisingly though, this competitive advantage is not sufficient for the parent strain to entirely outcompete and displace its bioluminescent derivative, which remains present in the gastrointestinal tract until clearance of both strains by the immune system. This suggests that there are sufficient niches within the gastrointestinal tract for the two strains to coexist.

It is important to note that in addition to light production, ICC180 differs from its non-bioluminescent parent strain ICC169 by lacking a putative site-specific DNA recombinase, disrupted by insertion of the lux operon. C. rodentium ICC180 was constructed by random transposon mutagenesis of ICC169 with a mini-Tn5 vector containing an unpromoted lux operon and kanamycin-resistance gene. Previous characterisation of the site of insertion of the lux operon suggested that the transposon had inserted within a homologue of the xylE gene. However, whole genome sequencing has revealed that this was incorrect and the lux operon has inserted at 5,212,273 bp, disrupting the coding region of the putative site-specific DNA recombinase. Whole genome sequencing also revealed that ICC180 differs from ICC169 by 2 non-synonymous SNPs, a single base pair insertion and a 90 bp deletion. It is unclear if these changes occurred during the process of transposon mutagenesis, and are merely ‘hitch-hikers,’ or after laboratory passage. The single base pair insertion revealed by sequencing is of a G residue at 3,326,092 bp which results in a frameshift mutation within a putative membrane transporter, while the 90 bp deletion is within the deoxyribose operon repressor gene deoR. The DeoR protein represses the deoCABD operon, which is involved in the catabolism of deoxyribonucleotides. One SNP is the substitution of an aspartic acid (D) for a glycine (G) at residue 471 of Cts1V, a Type 6 secretion system protein involved in ATP binding. The other SNP is the substitution of a glutamic acid (E) for a glycine (G) at residue 89 of the formate acetyletransferase 2 gene pflD, which is involved in carbon utilisation under anaerobic conditions. Modelling suggests that once mutated, residue 89 will be unable to make several key contacts, suggesting the function of PflD will be affected. As we have not introduced these genetic differences into the non-bioluminescent parent strain, we cannot be certain whether the fitness costs we observed are a result of any single or combination of these differences, or expression of the lux operon. In addition, at 54 kb the largest C. rodentium plasmid pCROD1 is dramatically altered in ICC180, missing 41 out of 60 of genes. This is in contrast to previous results which indicated that pCROD1 is entirely absent in ICC180 (Petty et al., 2011). We do not anticipate that the loss of a large part of this plasmid will have any significant impact however, as it has been shown that pCROD1 is frequently lost in C. rodentium, and that strains lacking pCROD1 do not show any attenuation of virulence in a C57BL/6 mouse model (Petty et al., 2011).

In conclusion, the bioluminescent C. rodentium strain ICC180 has a clear disadvantage when directly competed with its parent stain in mice. However, the fact that it reaches similar numbers, and causes similar pathology (Wiles et al., 2005; Wiles et al., 2006), during single infections is reassuring. Our phenotypic microarray data suggests that constitutive light expression is surprisingly neutral in C. rodentium and supports the view that bioluminescent versions of microbes can be used as a substitute for their non-bioluminescent parents, at least in theory. In reality, the actual fitness costs will likely depend on the host bacterial species, whether the lux operon is located on a multi-copy plasmid or integrated into the chromosome (and if chromosomal, the site of insertion of the operon), and the levels of expression of the lux genes.

Supplemental Information

Data S1 Raw data of bacterial counts, bioluminescence and growth in vitro and in vivo

Click here for additional data file.

Table S1 BIOLOG Phenotypic Microarray assays

Substrates and positions within the 96 well plates of the Biolog PM plates used in this study.

Click here for additional data file.

Table S2 Genes missing from pCROD1 of C. rodentium ICC180 9

List of genes missing from plasmid pCROD1 of C. rodentium ICC180, as determined by sequencing.

Click here for additional data file.

Figure S1 Elbow tests of phenotypic microarray array data to determine the number of clusters appropriate for k-means clustering

Data was analysed using the DuctApe software suite.

Click here for additional data file.

Figure S2 Phenotypic microarray (PM) growth curves of C. rodentium ICC180 and its non-bioluminescent parent strain ICC169 which are significantly different

Wildtype C. rodentium ICC169 (shown as purple lines) and its bioluminescent derivative ICC180 (shown as blue lines) were grown on two separate occasions using PM plates 1–20 (categorised by colour [see Key]). Differences between the growth of ICC169 and ICC180 in each individual well were analysed using the moderated t-test provided by limma. Wells in which the differences had an adjusted p-value of less than 0.5 (stringent cut-off) are shown.

Click here for additional data file.

Figure S3 Infection of larvae of the Greater Wax Moth Galleria mellonella with bioluminescent C. rodentium ICC180 can be visualised by luminometry

Groups of larvae (n = 10) of the Greater Wax Moth Galleria mellonella were infected with ∼108 CFU of C. rodentium ICC169 or ICC180 and monitored for bioluminescence using a plate luminometer. Data (medians with ranges) is presented from experiments performed on 3 separate occasions and is given as relative light units [RLU] waxworm−1.

Click here for additional data file.

Additional Information and Declarations

Competing Interests

Author Contributions

Animal Ethics

DNA Deposition

Data Availability

Siouxsie Wiles is an Academic Editor for PeerJ.

Hannah M. Read conceived and designed the experiments, performed the experiments, analyzed the data, wrote the paper, prepared figures and/or tables, reviewed drafts of the paper.

Grant Mills and Sarah Johnson performed the experiments, reviewed drafts of the paper.

Peter Tsai analyzed the data, prepared figures and/or tables, reviewed drafts of the paper.

James Dalton performed the experiments, analyzed the data, reviewed drafts of the paper.

Lars Barquist analyzed the data, contributed reagents/materials/analysis tools, reviewed drafts of the paper.

Cristin G. Print contributed reagents/materials/analysis tools, reviewed drafts of the paper.

Wayne M. Patrick analyzed the data, reviewed drafts of the paper.

Siouxsie Wiles conceived and designed the experiments, performed the experiments, analyzed the data, contributed reagents/materials/analysis tools, wrote the paper, prepared figures and/or tables, reviewed drafts of the paper.

The following information was supplied relating to ethical approvals (i.e., approving body and any reference numbers):

Experiments were performed in accordance with the New Zealand Animal Welfare Act (1999) and institutional guidelines provided by the University of Auckland Animal Ethics Committee, which reviewed and approved these experiments under applications R1003 and R1496.

The following information was supplied regarding the deposition of DNA sequences:

GenBank: SRP076686.

The following information was supplied regarding data availability:

Figshare: https://dx.doi.org/10.17608/k6.auckland.3407935.v1.

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
