# Peer review of "The in vitro and in vivo effects of constitutive light expression on a bioluminescent strain of the mouse enteropathogen Citrobacter rodentium"

_PeerJ, doi:10.7717/peerj.2130_

## Round 0.1 · original submission · Major Revisions

Reviewer 2 raises significant concerns about the experimental design and the conclusions drawn from the study. If you feel that you can addresses these specific concerns I am willing to consider a revised version of the manuscript. Substantial changes will be required to address these criticisms as well as the more minor points raised by both reviewers.

·

Basic reporting

This article is very well compiled and was a pleasure to review.

I have one minor issue and this is with the Figure legend of the final figure (7).This legend has too much technical description. it should state the difference between the groups and mention days which is not mentioned.

Experimental design

This is a unique piece of Research which addresses very pertinent questions in this area. Excellent thoughtful and reflectively designed set of experiments.

Validity of the findings

The data is very robust demonstrating sound rational findings which have been well discussed.

Reviewer 2 ·

Basic reporting

This is a potentially interesting and mostly well-written paper that addresses a somewhat esoteric, but worthy topic, namely, the impact of inserting a bioluminescence marker into the mouse pathogen, Citrobacter rodentium. Properties that were examined included growth in vitro under various conditions, and growth in two animal models: waxworms and mice - the latter being the natural host of C. rodentium.

Specific comments re the manuscript

1. This may be pedantic, but 'data' and 'media' are plural nouns. Hence, in lines 41, 144 and many other places in the manuscript plural verbs are required.

2. The abbreviation 'PCA' (line 146) should be explained.

3. In line 169, delete 'to' before 'retrospectively'.

4. Rewrite the sentence starting on line 216 by inserting something like 'media containing ' between 'on' and 'each'.

5. Rewrite the sentence starting on line 224, because it suggests that the anaesthetic was administered using a camera (see also the legend to Fig. 7).

6. At what temperature were the infected waxworms held? If this was not 37°C, temperature-regulated genes may have affected bacterial growth and virulence.

7. In line 343, use the abbreviation 'AUC' as you have introduced this previously.

8. In line 351, changed 'decreases' to 'decreased'.

9. In line 378, change 'was shown to have reduced mortality' to 'was shown to cause reduced mortality'

10. In line 384, delete 'have'

11. The quality of Fig. 1 is poor, and should be improved to make it more legible.

12. The details Fig. 2 can’t be seen. This figure doesn't add anything to what is provided in the text. It should be deleted or included with the supplementary data.

Experimental design

A major problem with this study in that the wild-type C. rodentium strain is not the ideal control for the test strain. This is because of differences between them other than bioluminescence, including some potentially significant SNPs and the presence of a kanamycin-resistance determinant in the test strain. Some of these problems are discussed by the authors in lines 438 ff.

A far better control would have been ICC180 (the bioluminescent test strain) with a disrupted lux gene. Without a suitable control strain, the findings of the study are difficult to interpret.

Validity of the findings

The authors conclude that the expression of bioluminescence had little effect on the fitness of C. rodentium in vitro and in waxworms, but the data they show are at odds with this conclusion. Specifically: compared to the wild-type strain, the test strain grew significantly less well in minimal medium (Fig. 4B) and significantly better in waxworms (Fig. 5D).

Although both strains were excreted in similar numbers in the faeces of orally-infected mice, the authors did not use any other measures of pathogenicity, such as colonic hyperplasia. Importantly, when the bioluminescent and wild-type strains were coadministered to mice, the wild-type significantly out-competed the test strain. Although the authors dismiss this finding, I strongly disagree with them, as a competitive index CI of <0.001, as seen here (Fig. 5E), is indicative of highly significant attenuation.

The authors claim that because the bioluminescent strain was still detectable in mice 15 hours after inoculation it wasn't really attenuated, does not negate my argument, as the CI had not reached a steady state when experiment was terminated (Fig. 5E). Moreover, in other studies with C. rodentium, investigators have typically measured the CI several days after inoculation and take CI values of <0.5 as being indicative of attenuation (see, e.g., Kelly et al. Infect Immun 2006; 74:2328-37).

In view of the data, the authors, need to soften their conclusions

Additional comments

No other comments

---

## Round 0.2 · accepted · Accept

While one of the reviewers had originally raised significant concerns about the choice of control strain for the study and the conclusions drawn from the data, I am happy to accept the revised version on the basis that the conclusions have been moderated significantly.